# Aerobic or Resistance Exercise for Improved Glycaemic Control and Pregnancy Outcomes in Women with Gestational Diabetes Mellitus: A Systematic Review

**DOI:** 10.3390/ijerph191710791

**Published:** 2022-08-30

**Authors:** Niamh Keating, Ciara Coveney, Fionnuala M. McAuliffe, Mary F. Higgins

**Affiliations:** 1UCD Perinatal Research Centre, National Maternity Hospital, University College Dublin, D02 YH21 Dublin, Ireland; 2Department of Midwifery, National Maternity Hospital, D02 YH21 Dublin, Ireland

**Keywords:** exercise, pregnancy, gestational diabetes, aerobic, strength

## Abstract

Exercise is often recommended in addition to diet and medication in the management of gestational diabetes mellitus (GDM). Our aim was to determine if strength training compared with aerobic exercise had an impact on glycaemic control, maternal and neonatal outcomes. The Cochrane library, Embase, PubMed, CINAHL, Medline, Google Scholar, and OpenGrey were searched. Over 758 pregnant women (mother-baby pairs) from 14 studies are included in this systematic review. Interventions ranged from cycling, aerobic exercises, walking, yoga, or combined aerobic and resistance exercises. Of the studies identified, none directly compared aerobic exercise with strength training. Half of the studies showed benefit in glycaemic control with additional exercise compared with usual physical activity. There was largely no impact on obstetric or neonatal outcomes. Studies on exercise in GDM have reiterated the safety of exercise in pregnancy and shown mixed effects on maternal glycaemic control, with no apparent impact on pregnancy outcomes. The heterogenicity of reported studies make it difficult to make specific recommendations on the optimum exercise modality for the management of GDM. The use of a core outcome set for GDM may improve reporting of studies on the role of exercise in its management.

## 1. Introduction

Gestational diabetes (GDM) is defined as new onset maternal hyperglycemia in pregnancy that resolves after birth [1]. Diabetes affects one in six pregnancies worldwide, of which the majority is GDM [2]. The rate of GDM diagnosed in pregnancy has increased over the last decade, with predictions of further increases worldwide [2,3]. GDM is associated with increased maternal and fetal morbidity and mortality [4,5]. Women with GDM are at a significantly increased risk of developing type 2 diabetes (T2DM) in later life [6]. Maintaining good glycaemic control can improve maternal and infant health outcomes [7]. 

Exercise is considered to be an important component of the management of GDM [8,9]. Outside of pregnancy, structured exercise is effective in improving diabetic control in patients with insulin resistance [10,11,12]. Pregnancy is a state of relative insulin resistance. Aerobic and resistance exercise have different metabolic effects and therefore both have a potential role in the treatment of insulin resistance seen in GDM [13,14,15].

Many studies have looked at the effect of exercise in the prevention of hypertensive disorders of pregnancy [16,17], pelvic or back pain [18,19,20], preterm birth [21,22], incontinence [23,24], and mental health disorders [25,26]. Most of the systematic reviews published on exercise in GDM have concentrated on exercise as a preventive measure to reduce the rate of GDM diagnosis [27,28,29]. 

We hypothesise that the type of exercise (either strength, aerobic, or combination) has an impact on glycaemic control and therefore health outcomes for both the mother and fetus. The aim of this systematic review was to update and consolidate the evidence on the effect of exercise modality on both glycaemic control and obstetric outcomes in patients with GDM.

## 2. Materials and Methods

This systematic review of randomised controlled trials (RCTs) and cohort and case control studies was carried out following the protocol described in the Preferred Reporting Items for Systematic Reviews and Meta-Analysis (PRISMA) statement [30]. Searches were initially performed from September to November 2019 and re-run in May 2021 prior to the final analyses so that further studies could be retrieved for inclusion. Electronic database searches were carried out using the following databases: Cochrane library, Embase, PubMed, Cochrane Central Register of Controlled Trial (CENTRAL), CINAHL, and Web of Science. Grey literature databases were also searched (Google Scholar and OpenGrey). Citation pearl indexing was also performed of previous systematic reviews, literature reviews, and guidelines. Studies were initially restricted to the last 15 years. There were no restrictions on language. 

### 2.1. Data Sources and Search Strategy

Two reviewers (NK and CC) performed a search to identify RCTs that studied the effect of exercise (aerobic/resistance/any exercise) on glycaemic control or obstetric outcomes in pregnancies affected by GDM. The search strategy followed the PICO framework, using key words, free text, and MeSH terms as appropriate and combining Boolean operators of (AND/OR/NOT/quotation marks/brackets): -**Participants:** women, pregnancy, gestational diabetes, hyperglycaemia, diabetes-**Intervention:** exercise, aerobic, resistance-**Comparison:** physical activity, aerobic, resistance, control-**Outcome:** medication (insulin, metformin), glycaemic control, maternal outcome

Inclusion criteria were as follows:Randomised controlled trialsGestational diabetes mellitusIntervention of resistance exercise or aerobic exercise alone or in combinationComparator or control of either resistance, aerobic or no exercise

Exclusion criteria were:Review or opinion articlesStudies without published resultsStudies involving women with pre-existing diabetes

The primary outcome was glycaemic control in women with GDM (defined as average blood glucose levels or use of insulin and insulin requirements where it was required to maintain euglycaemia). Secondary outcomes were obstetric outcomes including rate of caesarean birth, perineal trauma, or duration of labour.

### 2.2. Study Selection

All articles retrieved for review were imported to Endnote and duplicates were removed. Titles, abstracts, and full text were independently reviewed by two reviewers (N.K. and C.C.) with reference to a third reviewer (M.F.H.) if required. Inclusion and exclusion criteria are reported in Table 1. 

### 2.3. Data Extraction

Data extraction was independently performed by two researchers (N.K. and M.F.H.) using standardised electronic data extraction forms, saved on a shared drive after initial assessment, with reference to a third researcher (C.C.) as required. Missing data was to be requested from study authors. Data extraction was performed in accordance with the Cochrane Handbook for Systematic Review of Interventions [31]. Extracted data items included study reference details, study context, study design, study population, data analysis and methods, and research findings. 

### 2.4. Quality Assessment

Bias of randomised trials was assessed using the Cochrane Risk of Bias (RoB2) tool [32]. This study was registered with PROSPERO (2020 CRD42020161454) [33] and is reported following the PRISMA statement for systematic reviews [30]. 

## 3. Results

### 3.1. Literature Search

Flow of studies through the stages of identification, screening, eligibility, and final inclusion is shown in Figure 1. Fourteen randomised controlled studies were selected for inclusion in this systematic review [34,35,36,37,38,39,40,41,42,43,44,45,46,47,48]. Two of the papers [41,47] reported different outcomes from one trial and for the purpose of this manuscript are referred to as one paper.

### 3.2. Characteristics 

Characteristics of the fourteen studies are shown in Table 2 with risk of bias summary in Table 3 and results in Table 4. Study sizes ranged from six participants to 200 participants. In total, over 758 pregnant women (mother-baby pairs) are included in this systematic review. Length of intervention ranged from four to twelve weeks. Of the 14 papers analysed, five used aerobic exercise as the intervention, five used strength/flexibility-based exercise, two used a combined intervention, one did not specify the type of exercise used, and one compared strength or aerobic exercise with a control. Interventions ranged from supervised follow-up with a kinesiologist to cycling, aerobic exercises, walking, yoga, or combined aerobic and resistance exercises. 

No randomised studies were identified that directly compared aerobic to resistance exercise. Study summaries are shown in Table 4. One study compared aerobic or resistance exercise to controls, but the numbers were small, and the study was reported as a pilot study within a conference abstract, with plans to continue the study [36]. The most common comparison was “usual physical activity”. 

### 3.3. Fasting Glucose

Seven of the studies included fasting glucose as a primary outcome. Two papers [43,46] showed an improvement in fasting glucose in the intervention group. Youngwanichseta et al. [46] showed a statistically significant difference in fasting plasma glucose with intervention vs. control (post-test mean 83.39 ± 7.69 mmol/L vs. 87.85 mmol/L ± 7.94, *p* = 0.012). Jovanovic-Peterson [42] et al. found a reduction in fasting glucose with intervention vs. control (70.1 mmol/L ± 6.6 vs. 87.6 mmol/L ± 6.2, *p* < 0.001). The remaining showed no difference in fasting glucose with exercise. 

### 3.4. Postprandial Glucose

Seven of the papers looked at postprandial glucose as an outcome measure. Five papers [38,39,41,45,46] showed an improvement in postprandial glucose results in the intervention group. In Bo’s [37] study, postprandial glucose in the intervention group was 117.2 ± 16.5 mmol/L vs. 106.1 mmol/L ± 19.0 in the control group, giving an adjusted difference of minus 11.1 mmol/L (minus 16.1, minus 6.1, 95% CI, *p* < 0.001). In Halse’s [41] paper, there was a significant difference in postprandial glucose at breakfast, with lower levels in the intervention group compared with the control, (*p* = 0.046); postprandial glucose levels at dinner approached significance with lower levels in the exercise group, (*p* = 0.054), compared with the control. There was no difference in postprandial levels at lunch, (*p* = 0.312). Sklempe-Kokic et al. [45] showed an improvement in postprandial blood glucose in the intervention vs. control (mean 4.66 mmol/L ± 0.46 vs. 5.3 mmol/L ± 0.47, *p* < 0.001). In Youngwanicheta’s [46] paper, there was statistically significant difference in two hour postprandial glucose in the intervention vs. control (mean post-test glucose 103.67 mmol/L ± 9.93 vs. 114.36 mmol/L ± 10.15, *p* = 0.001). Brankston et al.’s [38] study showed a significant reduction in pooled post-meal glucose levels in the diet plus exercise group compared to diet alone with no difference in individual breakfast, lunch, and supper postprandial glucose levels. Two studies ([35,43]) showed no difference in postprandial blood sugar. 

### 3.5. Average Glucose

Five studies reported average glucose levels as an outcome. One study, by Halse [41], showed an improvement in mean postprandial glucose levels in the exercise group compared with the control (*p* = 0.004). The remaining four studies showed no difference [36,39,40,45].

### 3.6. HbA1C

Five papers recorded HbA1C as an outcome with four of these ([37,43,44,46]) showing a reduction in HbA1C with exercise intervention. In Bo’s [37] paper, HbA1C reduced from 4.9% ± 0.4 in the non-exercise group to 4.6% ± 0.5 in the exercise group with an adjusted difference of minus 0.3, *p* < 0.001. In Youngwanichseta’s [46] paper, there was a significant difference in post-test HbA1C with intervention vs. control (mean HbA1C 5.23% ± 0.22 vs. 5.68% ± 0.38, *p* = 0.03). Qadir [44] showed a significant difference in average HbA1c between the intervention and control groups (4.9% vs. 5.38%, *p* = 0.04). The paper by Ramos [43] is a conference abstract reporting the preliminary results of a pilot study. Post intervention mean HbA1c in the study group (*n* = 2) was 5.5% ± 0.4 compared with 6.3% ± 4.0 in the control group (*n* = 4). Halse [41] showed was no difference in HbA1C post study between intervention and control.

### 3.7. Insulin Use

Six papers studied the need for insulin treatment as an outcome, with one paper showing a reduction in the number of women requiring insulin with exercise intervention. There was a significant difference in De Barros’s paper [40] in the number of women who required insulin: 21.9% in the exercise group (7/32) vs. 56.3% (18/32) in the control group, *p* = 0.005. Five studies showed no difference [34,35,37,38,41]. 

Four papers reported changes in insulin dose. One paper (Brankston [38]) showed a reduction in insulin dose requirements with exercise intervention (diet plus exercise) vs. diet alone (0.22 units/kg ± 0.2 vs. 0.48 units/kg ± 0.3, *p* < 0.05). Halse’s paper [47] reported a clinical difference with a lower mean dose of insulin in the intervention (7 iu ± 1) group compared with the control (13 iu ± 1); the numbers of participants were too small to draw a statistically meaningful conclusion. Adam [34] and deBarros [40] showed no difference in mean dose of insulin between exercise intervention and control. Four studies reported latency to insulin requirement as an outcome. One paper [38] showed a statistically significant delay in starting insulin with intervention vs. control (3.71 weeks ± 3.1 vs. 1.11 weeks ± 0.8, 0 < 0.05), with three studies showing no difference [34,40,41].

### 3.8. Maternal Hypoglycaemia

There were no reports of maternal hypoglycaemia in the 14 papers analysed. 

### 3.9. Caesarean Section

Six papers reported rate of Caesarean section (CS) as an outcome. There was a significant difference in Awad’s [48] paper: the rate of CS was higher in the control group (63.4% (*n* = 19/30) compared with 16% (*n* = 5/30) in the exercise group, (*p*= 0.001)**.** Five studies showed no difference in CS rates between the exercise intervention and control groups [35,37,39,47]. 

### 3.10. Induction of Labour Rates and Labour Duration

Halse [41] was the only paper to report on induction of labour with no difference shown between intervention (55%) and control (42%), *p* > 0.05. They were also the only group to report on duration of labour with no difference between exercise intervention and control (445 min ± 309 vs. 348 min ± 187, *p* > 0.05)

### 3.11. Other Outcomes

None of the studies included reported on the use of Metformin, perineal trauma (third- or fourth-degree tears), or shoulder dystocia as an outcome.

## 4. Discussion 

There were no well-designed RCTs identified comparing aerobic and strength exercise in the management of GDM and therefore we are unable to comment on our primary outcome, but we were able to provide a narrative review for glycaemic control parameters and maternal outcomes related to exercise.

Pregnancy is a state of relative insulin resistance. Exercise has its insulin sensitising effects by increasing GLUT-4, increasing sensitivity GLUT-4 to insulin and increased glycogen synthase [49]. By this mechanism, uptake of glucose into muscles is increased. Aerobic and resistance exercise have different metabolic effects and therefore both have a potential role in the treatment of states of insulin resistance such as that seen in GDM. As skeletal muscle is the largest mass of insulin sensitive tissue, an increase in muscle mass through resistance training is associated with improved glycaemic control [13,14]. Aerobic exercise reduces visceral obesity which improves insulin sensitivity [15]. Aerobic exercise has been shown to be more beneficial in modulating insulin resistance and inflammatory cytokines in obese patients with T2DM when compared with resistance exercise [50]; however, it has been proposed that both methods likely incur benefit through their different mechanisms of action [51]. 

Previously, women had been advised against moderate exercise in pregnancy due to concerns about the risk of harm to the fetus including growth restriction and preterm birth. A systematic review [22] of over 2000 women found that in singleton, uncomplicated pregnancies, moderate exercise did not increase the risk of preterm birth or growth restriction. Other systematic reviews and meta-analyses have supported this finding [52,53], including showing a reduction in the rate of caesarean birth and gestational weight gain with exercise [54]. The American College of Obstetricians and Gynecologists (ACOG) [55] recommend 30 min or more of moderate exercise on most days for women without medical or obstetric risks. In one study [56], over half of women believed that weight-based exercise was unsafe in pregnancy. A lack of awareness of exercise in pregnancy guidelines among physicians has previously been shown [57]. As this review shows, there were no adverse effects reported in women with GDM, i.e., a pregnancy that may be considered “high risk”. A meta-analysis of women with risk factors such as obesity, hypertension, and GDM showed no adverse effect on the fetus with moderate intensity exercise [58]. An RCT largely comprised of previously sedentary women showed no increase in maternal or neonatal adverse effects with combined aerobic and resistance training [52]. Studies looking at the role of exercise in preventing GDM have shown conflicting results, with some studies showing a reduced incidence [59] and others showing no impact [60,61]. 

Of the studies analysed, eight out of the 14 showed some benefit with exercise intervention on parameters of glycaemic control, particularly lower fasting glucose [47,49], postprandial glucose [38,39,40,46,50,51,52], HbA1c [38,43,44,46], reduced need for insulin, and increased latency to starting insulin [38,40]. Different makers of glycaemic control were used which makes it difficult to compare results. Of the seven studies that included resistance training either alone or in combination, four of these demonstrated improved glycaemic control with intervention [38,40,45,46]. Two of these studies involving strength-based exercise as an intervention showed a reduction in the need for insulin or delayed the onset of starting insulin. The need for insulin treatment in GDM is an important clinical indicator of the degree of hyperglycaemia. Hyperglycaemia in pregnancy is associated with adverse outcomes [62] and pregnant women who require insulin as treatment are at a higher risk of adverse outcomes [63]. The impact of resistance exercise on eliminating the need for or delaying starting insulin in GDM is an important finding and more research supporting this would be helpful in advising patients. 

Of the five studies using aerobic exercise alone as an intervention, three [38,41,43] of these showed some improvement in glycaemic control. Of note, none of the studies included showed any negative effect of exercise. There is insufficient evidence to recommend one modality of exercise over another for the management of GDM and in practice, a combination of aerobic and strength is likely to show the most overall benefit in pregnancy as has been shown in the management of Type 2 DM [10]. 

The potential positive effect of exercise in GDM pregnancies on obstetric outcomes is largely theoretical modelling of the benefits of exercise in reducing hyperglycaemia. The studies presented have had mixed reports with many studies showing no difference with intervention. One study [48] showed a reduction in the CS rate among women with insulin controlled GDM who participated in a combined aerobic and strength exercise regime compared with standard care. When considering the global impact of GDM, it is notable that there are relatively few studies published in exercise as a treatment. Exercise is generally a free or low-cost intervention and is safe with few, if any, potential adverse effects and can be modified to all degrees of fitness and physical ability. Of the studies presented here, they appear to have been powered to look at variation in glycaemic control but were perhaps not large enough to report on maternal outcomes. 

A core outcome set (COS) for studies of GDM was published in 2020 [64] with 14 outcomes reported in the final set. All the papers included in this systematic review were published before the development of a core outcome set and much variation is seen in the outcomes measured; this reiterates the need for such a COS for consistency in reporting outcomes in publications. 

Several systematic reviews have been published on the topic of exercise for treatment for GDM. Cremona [65] looked at the effect of exercise modality on markers of insulin sensitivity in women with or at risk of GDM. They concluded that exercise sessions three times per week of either aerobic or strength training targeting major muscle groups could improve glycaemic control. Women with a high body mass index (BMI) at risk of GDM would also benefit; however, women with previous GDM pregnancies and a normal BMI do not appear to reduce their risk of GDM through exercise intervention. Huang [66] studied the effects of difference exercise modalities on glycaemic control alone but did not include five studies included in this review [34,36,43,44,48]. They performed a meta-analysis of nine RCTs involving 618 women which demonstrated high heterogenicity. The results showed that aerobic exercise reduced the fasting blood glucose, postprandial blood glucose, and HbA1C in patients with GDM compared to conventional treatment. They included Youngwanichestha et al.’s [46] study as an aerobic intervention although yoga is not generally considered aerobic as the intention is not to increase the heart rate. The dosage of insulin was reduced in the resistance exercise group compared with conventional treatment; however, this was based on two studies [38,40]. The combination of aerobic and strength exercise compared with conventional treatment reduced postprandial glucose levels, but this was based on only one study [45]. Brown [67] performed a systematic review of 11 studies involving 638 women and concluded that the evidence is poorly reported and confounded by the variety of exercise interventions. Harrison [68] studied eight studies with self-monitored postprandial glucose as the primary outcome and concluded that exercise performed at moderate intensity a minimum of three times a week appears to be effective in managing GDM, although again this review was limited by the limited number of studies, heterogenicity in study design, and reporting of outcomes. Bgengski [69] included the same eight studies in their systematic review but only studied the effect of exercise on fasting plasma glucose as their primary outcome, concluding there was no difference between exercise and physical activity counselling compared with standard care on fasting plasma glucose levels. There was no difference on secondary outcomes which were macrosomia, preterm birth, CS, GA at delivery, and birth weight. Allehandan [70] included eight studies in their systematic review and concluded that diet plus exercise for women with GDM lowered fasting and postprandial glucose levels compared with diet alone. 

Only two of the systematic reviews in this area [69,70] included both maternal outcomes as well as glycaemic control. While glycaemic control is an important predictor of morbidity in GDM, it is crucial that obstetric and neonatal outcomes are reported, as ultimately this is the information that will be used to counsel patients and inform guidelines. Some abstracts from the grey literature included in this analysis have not been reported in previous systematic reviews. Given the lack of large RCTs, it is important that smaller studies such as those published at conferences be included in systematic reviews to allow a complete review of the available literature. 

### Strengths and Limitations

A strength of this study is the inclusion of studies not included in previous systematic reviews and inclusion of obstetric outcomes which have often been excluded previously. As with previous systematic reviews, it is limited by the small number of studies and the lack of large RCTs. We could not identify a study that directly compared strength training with aerobic exercise in the management of GDM and therefore cannot answer our primary research question.

## 5. Conclusions 

Studies on exercise for the management of GDM have shown mixed effects on maternal glycaemic control, with disappointingly no apparent impact on pregnancy outcomes. 

Despite exercise being integral to GDM management, the evidence bases for medium to long term benefit remains lacking. This is an important topic for discussion that reveals a paucity of data to guide healthcare providers in recommendations on exercise modality for the management of GDM. Further well designed large RCTs using a core outcome set are needed to determine the most efficient way to use exercise to treat GDM. Ideally, future studies would continue into the postpartum period to determine the effect of resistance or aerobic exercise on long term progression to T2DM following a pregnancy with GDM. 

The development of a COS for diabetes will improve reporting in studies on the role of exercise in GDM. The heterogenicity of reported studies make it difficult to make specific recommendations on the optimum exercise regime. Given the different effects aerobic and strength training have on glucose metabolism, it is plausible that a combination of both modalities is useful in controlling GDM.

## Figures and Tables

**Figure 1 ijerph-19-10791-f001:**
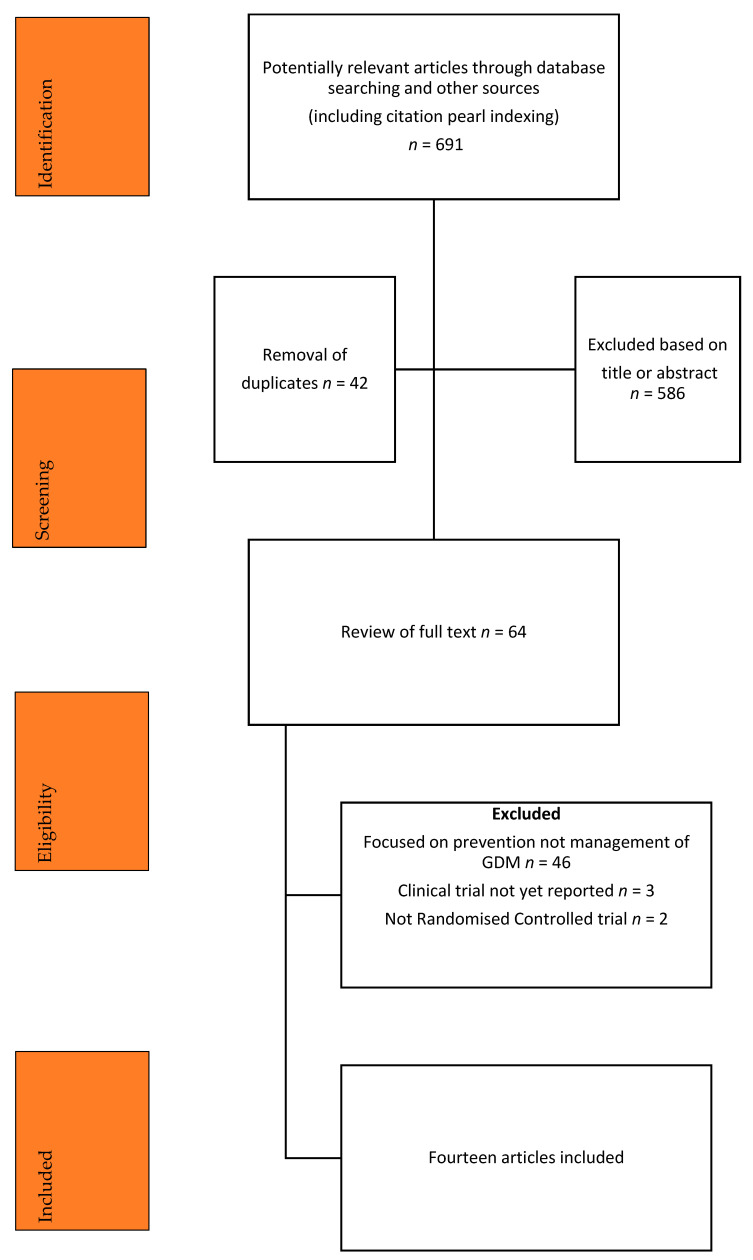
Flow of studies through the systematic searching of the literature.

**Table 1 ijerph-19-10791-t001:** PICOS criteria for inclusion and exclusion of studies in this Systematic Review.

Parameter	Inclusion	Exclusion
**Population/** **participant**	Women Pregnancy Gestational diabetes Hyperglycaemia Diabetes in pregnancy Treatment of Gestational Diabetes	Pre-gestational Diabetes Type 1 Diabetes Type 2 Diabetes Prevention of Gestational Diabetes
**Intervention**	Exercise, aerobic, resistance Lasting at least two weeks	
**Comparison**	Normal physical activity Aerobic exercise Resistance Control Lasting at least two weeks	
**Outcome**	Glycaemic control Maternal outcomes	
**Study Design**	Randomised controlled trial No language restriction Published paper Published abstract	Case report Case-control study Cohort study Commentary Guideline

**Table 2 ijerph-19-10791-t002:** Characteristics of studies included: Sample characteristics, Duration of interventions, Intervention (Exercise) Characteristics and Outcomes.

Author of RCT; Year Published Country	Sample Characteristics	Duration of Intervention	Intervention (Exercise) Characteristics	Outcomes
Adam 2014 [34]	Control *n* − 4 Intervention *n* = 39	Duration of pregnancy	Standard counselling for physical activity Supervised individuals follow up with kinesiologist	Addition of insulin Mean dose of insulin Time to start insulin Weight gain
Avery 1997 [35] United States	Control *n* = 14 GDM diagnosis 26.3 ± 8 weeks Intervention *n* = 15 GDM diagnosis 28.7 ± 3 weeks	Six weeks	Usual physical activity 30 min supervised cycling with 30 min unsupervised walking	Addition of insulin Caesarean birth Hypertensive disorders Maternal weight gain
Awad 2019 [48] Egypt	Control *n* = 30 Diet plus insulin Intervention *n* = 30 Moderate intensity aerobic and circuit resistance exercise 3–4 times/week in addition to diet plus insulin	24 weeks’ gestation until delivery	Diet plus insulin alone compared with combined strength and aerobic exercise plus diet plus insulin	Mode of delivery
Bambicini 2012 [36]	Control *n* = 6 Intervention (aerobic) *n* = 6 Intervention (resistance) *n* = 5	Duration of pregnancy	Seated listening to explanations about exercise Aerobic or resistance exercises	Mean glucose immediately after session and one hour later
Bo 2014 [37] Italy	Control *n* = 99 GDM diagnosis 24–26 weeks Intervention *n* = 101 GDM diagnosis 24–26 weeks	12–14 weeks	Not applicable Twenty minutes of unsupervised brisk walking seven times a week	Addition of insulin Caesarean birth Glycaemic control
Brankston 2004 [38] Canada	Control *n* = 16 GDM diagnosis: not reported Intervention *n* = 16 GDM diagnosis: not reported	Eight weeks	Usual physical activity Resistance exercise on circuit: supervised for three sessions then supervised for three sessions per week	Additional medications Glycaemic control
Bung 1991 [39] United States	Control *n* = 17 GDM diagnosis: 30.3 ± 2 weeks Intervention *n* = 17 GDM diagnosis: 30.3 ± 1.9 weeks Note: control was diet and insulin; diagnosis of GDM was persistent fasting glucose >5.88 mM but <7.22 mM and “failed diet therapy for a week”	Remainder of the pregnancy	Standard care Supervised in exercise laboratory: 45 min with breaks on recumbent bicycle	Adherence to intervention Caesarean birth Hypoglycaemia Glycaemic control
De Barros 2010 [40] Brazil	Control *n* = 32 GDM Diagnosis 27.5 ± 3 weeks Intervention *n* = 32 GDM Diagnosis 28.4 ± 2.5 weeks	Eight weeks	Usual physical activity Resistance exercise (two supervised and one unsupervised) for 30–40 min	Additional medications Caesarean birth Weight gain
Halse 2014 [41] Australia	Control *n* = 20 GDM Diagnosis 28.8 ± 1 week Intervention *n* = 20 GDM Diagnosis 28.9 ± 1 week	Six weeks	Usual physical activity Home cycle ergometer supervised three times a week and unsupervised for two sessions a week	Additional medications Caesarean birth Induction of labour Patient views Weight gain
Jovanovic-Peterson 1989 [42] United States	Control *n* = 9 GDM diagnosis at 28 weeks Intervention *n* = 10 GDM diagnosis at 28 weeks	Six weeks	Usual physical activity Aerobic exercise: 20 min for three times a week, supervised, using ergometer	Additional medications Adherence to intervention Hypertensive disorders Glycaemic control
Qadir 2018 [44] Singapore	Control *n* = 5 GDM “newly diagnosed” Intervention *n* = 5 GDM “newly diagnosed”	Eight weeks	Usual physical activity measured by pedometer Patient education and structured exercise class once a week Usual physical activity measured by pedometer	Average daily steps Glycaemic control
Ramos 2015 [43]	Control *n* = 4 Intervention *n* = 2	Ten weeks	50 min stretching and relaxation once a week 50 min aerobic session three times a week	Mean HbA1c Homeostatic model assessment (HOMA)
Sklempe Kocic 2018 [45] Croatia	Control *n* = 20 GDM diagnosis: 20.8 ± 6 weeks Intervention *n* = 18 GDM diagnosis 22.2 ± 6 weeks	Six weeks	Usual physical activity Combined aerobic and resistance exercise (two supervised sessions) plus seven sessions of unsupervised walking	Additional medications Caesarean Birth Glycaemic control Weight gain
Youngwanichsetha 2014 [46] Thailand	Control *n* = 85 GDM diagnosis: 24–30 weeks Intervention *n* = 85 GDM diagnosis: 24–30 weeks	Eight weeks	Not applicable Fifteen to twenty minutes of supervised yoga five times a week	Glycaemic control

**Table 3 ijerph-19-10791-t003:** Risk of Bias Summary.

Author of RCT; Year Published	Selection Bias (Random Sequence Generation)	Selection Bias (Allocation Concealment)	Performance Bias (Double Blinding)	Detection Bias (Blinding of Outcome Assessment)	Attrition Bias (Incomplete Outcome Data)	Reporting Bias (Selective Reporting)	Other Bias
Adam 2014 [34]	Unclear	Unclear	Unclear	Unclear	Unclear	Unclear	Unclear
Avery 1997 [35]	Low	Unclear	High	Unclear	High	Unclear	Low
Awad 2019 [48]	Unclear	Unclear	High	Unclear	Unclear	Unclear	Low
Bambicini 2012 [36]	Unclear	Unclear	Unclear	Unclear	Unclear	Unclear	Unclear
Bo 2014 [37]	Unclear	Low	High	Low	Low	Low	Low
Brankston 2004 [38]	Low	Low	High	High	Unclear	High	Unclear
Bung 1991 [39]	Unclear	Unclear	Unclear	Unclear	High	Unclear	Unclear
De Barros 2010 [40]	Low	Low	High	High	Low	Unclear	Low
Halse 2014 [41]	Unclear	Low	High	High	High	High	Low
Jovanovic-Peterson 1989 [42]	Low	Unclear	High	Unclear	Low	High	High
Qadir [44]	Unclear	Unclear	Unclear	Unclear	Unclear	Unclear	Unclear
Ramos 2015 [43]	Unclear	Unclear	Unclear	Unclear	Unclear	Unclear	Unclear
Sklempe Kocic 2018 [45]	Low	Unclear	High	Low	Low	High	Low
Youngwanichsetha 2014 [46]	Unclear	Unclear	High	Low	Low	High	Low

**Table 4 ijerph-19-10791-t004:** Results.

Paper	Intervention	Sample Characteristics	Main Outcome	Findings
Adam et al. [34]	Standard counselling for physical activity compared with supervised individual follow up with kinesiologist	Control *n* − 40 Intervention *n* = 39	Addition of insulin Mean dose of insulin Time to start insulin Weight gain	No difference
Avery et al. [35]	Usual physical activity compared with supervised cycling and unsupervised walking	Control *n* = 14 GDM diagnosis 26.3 ± 8 weeks Intervention *n* = 15 GDM diagnosis 28.7 ± 3 weeks	Addition of insulin Apgar < 7 at 1 min Apgar < 7 at 5 min Birthweight Caesarean birth Gestation at birth Hypertensive disorders Maternal weight gain	No difference in insulin requirement No difference in CS, hypertensive disorders
Awad et al. [48]	Diet plus insulin alone compared with combined strength and aerobic exercise plus diet plus insulin	Control *n* = 30 Diet plus insulin Intervention = 30 Moderate intensity aerobic and circuit resistance exercise 3–4 times/week in addition to diet plus insulin	Mode of delivery	Reduced CS rate in control group
Bambicini et al. [36]	Explanation about exercise compared with aerobic or strength-based exercise	Control *n* = 6 Intervention (aerobic) *n* = 6 Intervention (resistance) *n* = 5	Mean glucose immediately after session and one hour later	No difference
Bo et al. [37]	Twenty minutes of brisk walking 7 times per week	Control *n* = 99 GDM diagnosis 24–26 weeks Intervention *n* = 101 GDM diagnosis 24–26 weeks	Addition of insulin Caesarean birth Glycaemic control	No difference in insulin requirements, CS, macrosomia, fasting glucose Reduction in postprandial glucose and HbA1C with intervention
Brankston et al. [38]	Usual physical activity Resistance exercise on circuit: supervised for three sessions then supervised for three sessions per week	Control *n* = 16 GDM diagnosis: not reported Intervention *n* = 16 GDM diagnosis: not reported	Additional medications Glycaemic control	Increased latency to insulin treatment in intervention group No difference in number of women requiring insulin or the dose used No difference in fasting or postprandial glucose levels
Bung et al. [39]	Standard care Supervised 45 min session on recumbent bicycle	Control *n* = 17 GDM diagnosis: 30.3 ± 2 weeks Intervention *n* = 17 GDM diagnosis: 30.3 ± 1.9 weeks Note: control was diet and insulin; diagnosis of GDM was persistent fasting glucose >5.88 mM but <7.22 mM and “failed diet therapy for a week”	Adherence intervention Caesarean birth Hypoglycaemia Glycaemic control	No difference in CS, average glucose levels
De Barros et al. [40]	Usual physical activity Resistance exercise (two supervised and one unsupervised) for 30–40 min	Control *n* = 32 GDM Diagnosis 27.5 ± 3 weeks Intervention *n* = 32 GDM Diagnosis 28.4 ± 2.5 weeks	Additional medications Caesarean birth Weight gain	No difference in latency to use of insulin or dose required Reduction in number of women requiring insulin No difference in caesarean section
Halse et al. [41]	Usual physical activity Home cycle ergometer supervised three times a week and unsupervised for two sessions a week	Control *n* = 20 GDM Diagnosis 28.8 ± 1 week Intervention *n* = 20 GDM Diagnosis 28.9 ± 1 week	Compliance, maternal attitudes to exercise, aerobic fitness, onset of labour, mode of delivery, duration of labour	No difference in maternal obstetric outcomes, improved fitness, attitude, and exercise intention
Halse et al. [47]	Usual physical activity Home cycle ergometer supervised three times a week and unsupervised for two sessions a week	Control *n* = 9 GDM diagnosis at 28 weeks Intervention *n* = 10 GDM diagnosis at 28 weeks	Glycaemic control	No difference in fasting glucose, HbA1C, insulin use, dose, or latency to starting insulin Improved post prandial glucose and average glucose
Jovanovic-Peterson et al. [42]	Usual physical activity Aerobic exercise: 20 min for three times a week, supervised, using ergometer	Control *n* = 5 GDM “newly diagnosed” Intervention *n* = 5 GDM “newly diagnosed”	Additional medications Adherence to intervention Hypertensive disorders Glycaemic control	Improved fasting glucose with intervention No difference in obstetric or maternal outcomes
Qadir et al. [44]	Usual physical activity measured by pedometer Patient education and structured exercise class once a week Usual physical activity measured by pedometer	Control *n* = 4 Intervention *n* = 2	Average daily steps Glycaemic control	No difference in glycaemic control
Ramos et al. [43]	50 min stretching and relaxation once a week 50 min aerobic session three times a week	Control *n* = 20 GDM diagnosis: 20.8 ± 6 weeks Intervention *n* = 18 GDM diagnosis 22.2 ± 6 weeks		Improved HbA1C in intervention
Sklempe Kokic [45]	Usual physical activity Combined aerobic and resistance exercise (two supervised sessions) plus seven sessions of unsupervised walking	Control *n* = 85 GDM diagnosis: 24–30 weeks Intervention *n* = 85 GDM diagnosis: 24–30 weeks	Additional medications Caesarean Birth Glycaemic control Weight gain	Improved postprandial glucose with intervention, no difference in maternal obstetric outcomes
Youngwanichsetha et al. [46]	Not applicable Fifteen to twenty minutes of supervised yoga five times a week		Glycaemic control	Improved fasting glucose, post prandial glucose, HbA1C with intervention

## Data Availability

Not applicable.

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
