# Peer review of "Aerobic or Resistance Exercise for Improved Glycaemic Control and Pregnancy Outcomes in Women with Gestational Diabetes Mellitus: A Systematic Review"

_ijerph, 2022, doi:10.3390/ijerph191710791_

Round 1
Reviewer 1 Report
I think this is an important paper but the main conclusion has been missed. Essentially, the practising clinician will benefit from having it stressed that exercise is safe in pregnancy and the type does not matter. These are clear practical considerations that arise from this work.
This is covered to some extent in the discussion but could be stressed. The points should also be added to the abstract.
On page 9 be consistent and do not use mg/dl and mmol/l in the same section use mmol/l as SI units.
Author Response
Thank you very much for taking the time to review this paper, your input is very much appreciated. We have made the following changes based on your advice:
- added further to the discussion on the point that exercise is safe in pregnancy
- added to the abstract on the point that exercise is safe in pregnancy
-used mmol/L rather than both mg/dl and mmol/L
Thank you again
Reviewer 2 Report
This systematic review of aerobic versus resistance exercise in gestational diabetes highlights an area with the need for further research in particular large randomised trials using core outcome sets.
It has been well performed and written and I have only minor comments for improvements. I think it would be helpful for the reader if the number of participants was added into table 4 (results) to make it easier to interpret how likely the lack of difference in outcomes is related to inefficacy versus lack of power. There is a typo on page 5 line 114 where table 3 was referred to as reporting outcomes (rather than risk of bias summary).
Author Response
Thank you very much for the time you have taken to review this paper. We are very grateful. We have changed the error in Page 5 and included a column with the numbers in the studies in Table 4.
We appreciate the time you have taken to review this paper.